# Neuronal populations in the occipital cortex of the blind synchronize to the temporal dynamics of speech

Markus Johannes Van Ackeren[1], Francesca M Barbero[2,3], Stefania Mattioni[1], Roberto Bottini[1], Olivier Collignon[1,2,3]*

[1]Center for Mind/Brain Studies, University of Trento, Trento, Italy; [2]Institute of research in Psychology, University of Louvain, Louvain, Belgium; [3]Institute of Neuroscience, University of Louvain, Louvain, Belgium

**Abstract** The occipital cortex of early blind individuals (EB) activates during speech processing, challenging the notion of a hard-wired neurobiology of language. But, at what stage of speech processing do occipital regions participate in EB? Here we demonstrate that parieto-occipital regions in EB enhance their synchronization to acoustic fluctuations in human speech in the theta-range (corresponding to syllabic rate), irrespective of speech intelligibility. Crucially, enhanced synchronization to the intelligibility of speech was selectively observed in primary visual cortex in EB, suggesting that this region is at the interface between speech perception and comprehension. Moreover, EB showed overall enhanced functional connectivity between temporal and occipital cortices that are sensitive to speech intelligibility and altered directionality when compared to the sighted group. These findings suggest that the occipital cortex of the blind adopts an architecture that allows the tracking of speech material, and therefore does not fully abstract from the reorganized sensory inputs it receives.

DOI: https://doi.org/10.7554/eLife.31640.001

*For correspondence:
olivier.collignon@uclouvain.be

**Competing interests:** The authors declare that no competing interests exist.

## Introduction

The human cortex comprises a number of specialized units, functionally tuned to specific types of information. How this functional architecture emerges, persists, and develops throughout a person's life are among the most challenging and exciting questions in neuroscience research. Although there is little debate that both genetic and environmental influences affect brain development, it is currently not known how these two factors shape the functional architecture of the cortex. A key topic in this debate is the organization of the human language system. Language is commonly thought to engage a well-known network of regions around the lateral sulcus. The consistency of this functional mapping across individuals and its presence early in development are remarkable, and often used to argue that the neurobiological organization of the human language system is the result of innate constraints (*Dehaene-Lambertz et al., 2006*; *Berwick et al., 2013*). Does the existence of a highly consistent set of regions for language acquisition and processing imply that this network is 'hard-wired' and immutable to experience? Strong nativist theories for linguistic innateness leave little room for plasticity due to experience (*Bates, 1999*), suggesting that we should conceive 'the growth of language as analogous to the development of a bodily organ' (*Chomsky, 1976*, p.11). However, studies in infants born with extensive damage to cortical regions that are typically involved in language processing may develop normal language abilities, thereby demonstrating that the language network is subject to reorganization (*Bates, 2005*). Perhaps the most intriguing demonstrations to show that the neurobiology of language is susceptible to change due to experience come from studies showing functional selectivity to language in primary and secondary 'visual' areas in

**eLife digest** Scientists once thought that certain parts of the brain were hard-wired to process information from specific senses or to perform specific tasks. For example, some had concluded that language processing is built into certain parts of the brain, because the way the brain responds to language is remarkably similar in different people even from very early on in life. Yet other studies with individuals who were born blind emphasize that experience also shapes the way the brain works. In people who are born blind, parts of the brain that typically interpret visual information in sighted people are often put to other uses.

Now, van Ackeren et al. show that people who became blind early in life are able to repurpose parts of the brain that are more typically used for vision to understand spoken language instead. A technique called magnetoencephalography was used to map how different parts of the brain respond when both people with sight and those who are blind listen to recordings of someone talking. In some of the experiments, the speech was distorted, making it unintelligible. In both groups, areas of the brain known to process sound information showed patterns of activity that match the rhythms present in the speech. The group with blindness also showed similar activity in parts of the brain usually used to process visual information, and even more so when they were exposed to intelligible speech.

The experiments show that brain efficiently reshapes to adapt to a world with no visual input. It may do this by making use of connections that already exist between the auditory and visual brain centers. For instance, very young children use these connections to link what they hear to the lip movements of adults. Future studies are needed to determine if individuals whose ability to see is restored would be able to process the visual information or if the adaptation of the visual processing parts of the brain to help understand speech would interfere with their sight.

DOI: https://doi.org/10.7554/eLife.31640.002

congenitally blind individuals (*Röder et al., 2002*; *Burton, 2003*; *Amedi et al., 2004*; *Bedny et al., 2011*; *Arnaud et al., 2013*). Such reorganization of the language network is particularly fascinating because it arises in the absence of injury to the core language network (*Bates, 2005*; *Atilgan et al., 2017*).

However, the level at which the occipital cortex is involved in speech representation in the early blind (EB), remains poorly understood. Speech comprehension requires that the brain extracts meaning from the acoustic features of sounds (*de Heer et al., 2017*). Although several neuroimaging studies have yielded valuable insights about the processing of speech in EB adults (*Arnaud et al., 2013*; *Bedny et al., 2011*; *Büchel, 2003*; *Lane et al., 2015*; *Röder et al., 2002*) and infants (*Bedny et al., 2015*), these methods do not adequately capture the fast and continuous nature of speech processing. Because speech unfolds over time, understanding spoken language relies on the ability to track the incoming acoustic signal in near real-time (*Peelle and Davis, 2012*). Indeed, speech is a fluctuating acoustic signal that rhythmically excites neuronal populations in the brain (*Poeppel et al., 2008*; *Gross et al., 2013*; *Peelle et al., 2013*). Several studies have demonstrated that neuronal populations in auditory areas entrain to the acoustic fluctuations that are present in human speech around the syllabic rate (*Luo and Poeppel, 2007*; *Kayser et al., 2009*; *Szymanski et al., 2011*; *Zoefel and VanRullen, 2015*). It has therefore been suggested that entrainment reflects a key mechanism underlying hearing by facilitating the parsing of individual syllables through adjusting the sensory gain relative to fluctuations in the acoustic energy (*Giraud and Poeppel, 2012*; *Peelle and Davis, 2012*; *Ding and Simon, 2014*). Crucially, because some regions that track the specific acoustic rhythm of speech are sensitive to speech intelligibility, neural synchronization is not only driven by changes in the acoustic cue of the auditory stimuli, but also reflects cortical encoding and processing of the auditory signal (*Peelle et al., 2013*; *Ding and Simon, 2014*). Speech tracking is therefore an invaluable tool to probe regions that interface speech perception and comprehension (*Poeppel et al., 2008*; *Gross et al., 2013*; *Peelle et al., 2013*).

Does the occipital cortex of EB people synchronize to speech rhythm? Is this putative synchronization of neural activity to speech influenced by comprehension? Addressing these questions would provide novel insights into the functional organization of speech processing rooted in the occipital

cortex of EB people. In the current study, we investigated whether neuronal populations in blind occipital cortex synchronize to rhythmic dynamics of speech, by relating the amplitude fluctuations in speech to electromagnetic dynamics recorded from the participant's brain. To this end, we quantified the local brain activity and directed connectivity in a group of early blind (EB; n = 17) and sighted individuals (SI; n = 16) using magnetoencephalography (MEG) while participants listened to short narrations from audiobooks. If the occipital cortex of the blind entrains to speech rhythms, this will support the idea that this region processes low-level acoustic features relevant for understanding language. We further tested whether the putative synchronization of occipital responses in EB people benefits from linguistic information or only relates to acoustic information. To separate linguistic and acoustic processes, we relied on a noise-vocoding manipulation. This method spectrally distorted the speech signal in order to impair intelligibility gradually, but systematically preserves the slow amplitude fluctuations responsible for speech rhythm (*Shannon et al., 1995*; *Peelle and Davis, 2012*). Furthermore, to go beyond differences in the local encoding of speech rhythm in the occipital cortex, we also investigated whether the connectivity between occipital and temporal regions sensitive to speech comprehension is altered in early blindness.

## Results

### Story comprehension

Participants listened to either natural speech segments (nat-condition) or to altered vocoded versions of these segments (see Materials and methods for details). In the 8-channel vocoded condition, the voice of the speaker is highly distorted but the intelligibility is unperturbed. By contrast, in the 1-channel vocoded condition, the speech is entirely unintelligible. After listening to each speech segment, participants were provided with a short statement about the segment, and asked to indicate whether the statement was true or false. Behavioral performance on these comprehension statements was analyzed using linear mixed-effects models with maximum likelihood estimation. This method is a linear regression that takes into account dependencies in the data, as present in repeated measures designs. Blindness and intelligibility were included as fixed effects, while subject was modeled as a random effect. Intelligibility was nested in subjects. Intelligibility had a significant effect on story comprehension ($\chi(2)$=110.7, p<0.001). The effect of blindness and the interaction between intelligibility and blindness were non-significant ($\chi(1)$=1.14, p=0.286 and $\chi(2)$=0.24, p=0.889). Orthogonal contrasts demonstrated that speech comprehension was stronger in the nat and 8-channel condition versus the 1-channel condition (b = 0.78, $t(62)$=14.43, p<0.001, r = 0.88). There was no difference between the nat and the 8-channel condition (b = 0.02, $t(62)$=1.39, p=0.17). Thus, speech intelligibility was reduced in the 1-channel, but not the 8-channel vocoded condition (*Figure 1D*). The lack of effect for the factor blindness suggests that there is no evidence for a potential difference in comprehension, attention or motivation between groups. Descriptive statistics of the group, and condition means are depicted in *Table 1*.

### Tracking intelligible speech in blind and sighted temporal cortex

The overall coherence spectrum, which highlights the relationship between the amplitude envelope of speech and the signal recorded from the brain, was maximal over temporal sensors between 6 Hz and 7 Hz (*Figure 1E*). This first analysis was performed on the combined dataset, and hence is not prone to bias or circularity for subsequent analyses, targeting group differences. The peak in the current study is slightly higher than those reported in previous studies (*Gross et al., 2013*). A likely explanation for this shift is the difference in syllabic rate between English (~6.2 Hz), used in previous studies, and Italian (~7 Hz) (*Pellegrino et al., 2011*). The syllabic rate is the main carrier of amplitude fluctuations in speech, and thus is most prone to reset oscillatory activity.

To capture the temporal scale of cerebro-acoustic coherence effects (6–7 Hz) optimally, and to achieve a robust source estimate, source reconstruction was performed on two separate frequency windows (6 ± 2, and 7 ± 2 Hz) for each subject and condition. The two source images were averaged for subsequent analysis, yielding a single image representing the frequency range 4–9 Hz. By combining the two frequency bands, we acquire a source estimate that emphasizes the center of our frequency band of interest (6–7 Hz) and tapers off towards the edges. This source estimate optimally represents the coherence spectrum observed in sensor space. The choice of the center

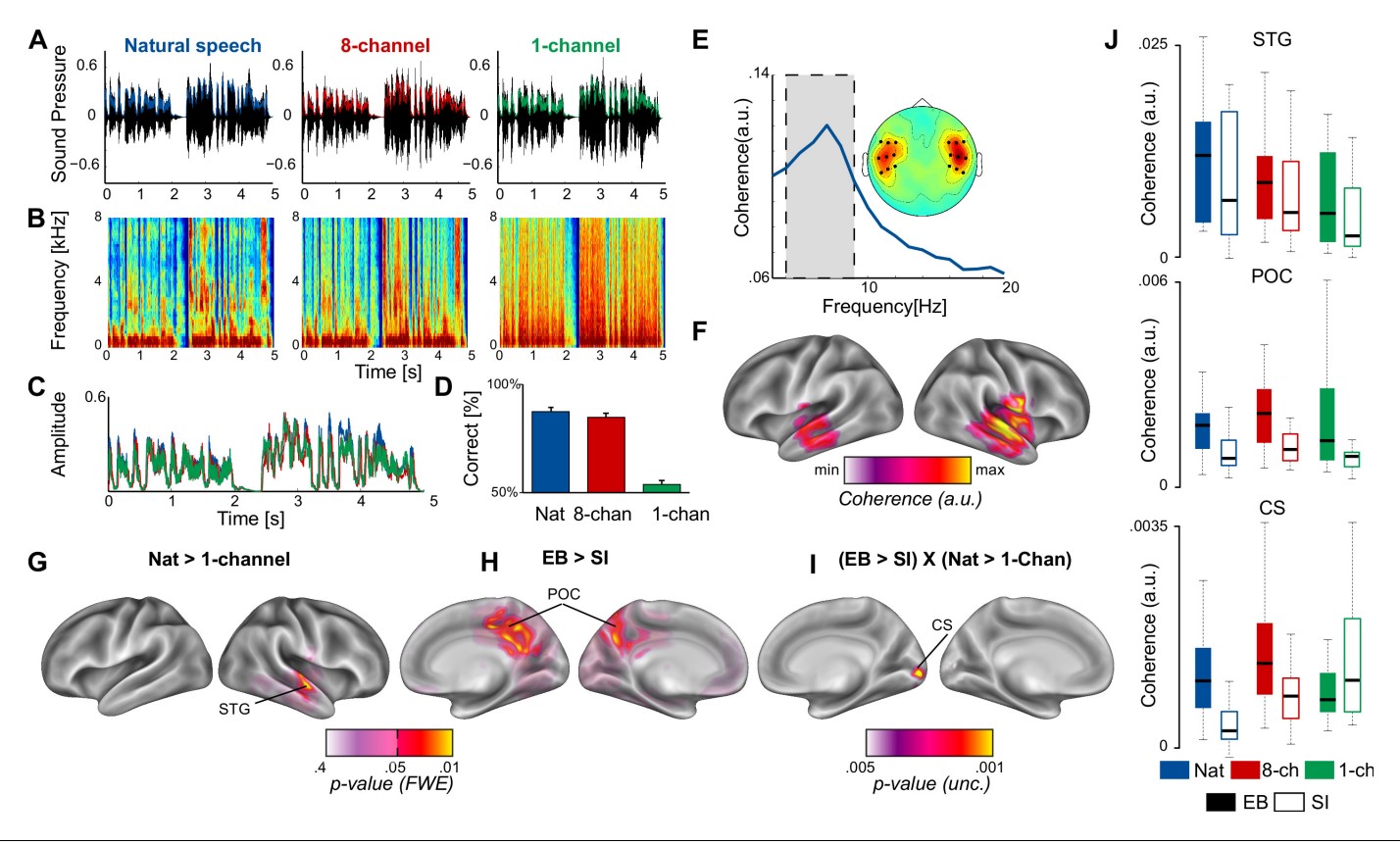

**Figure 1.** Cerebro-acoustic coherence in EB and SI. (**A**) Segments from each condition (nat, 8-channel, and 1-channel) in the time domain (see also *Video 1*). Overlaid is the amplitude envelope of each condition. (**B**) Spectrogram of the speech samples from (**A**). The effect of spectral vocoding on the fine spectral detail in all three conditions. (**C**) The superimposed amplitude envelopes of the samples for each condition are highly similar, despite distortions in the fine spectral detail. (**D**) Behavioral performance on the comprehension statements reveals that comprehension is unperturbed in the nat and 8-channel conditions, whereas the 1-channel condition elicits chance performance. (**E**) Coherence spectrogram extracted from bilateral temporal sensors reveals a peak in cerebro-acoustic coherence at 7 Hz, across groups and conditions. The shaded area depicts the frequency range of interest in the current study. The topography shows the spatial extent of coherence values in the range 4–9 Hz. Enhanced coherence is observed selectively in bilateral temporal sensors. (**F**) Source reconstruction of the raw coherence confirms that cerebro-acoustic coherence is strongest in the vicinity of auditory cortex, bilaterally, extending into superior and middle temporal gyrus. (**G**) Statistically thresholded maps for the contrast between natural and 1-channel vocoded speech show an effect of intelligibility (p<0.05, FWE-corrected) in right STG. (**H**) Enhanced envelope tracking is observed for EB versus SI in a bilateral parieto-occipital network along the medial wall centered on Precuneus (p<0.05, FWE-corrected). (**I**) The statistical map shows the interaction effect between blindness and intelligibility: Early blind (EB) individuals show enhanced synchronization during intelligible (nat) versus non-intelligible speech (1-channel) as compared to SI in right calcarine sulcus (CS) (p<0.005, uncorrected). (**J**) Boxplots for three regions identified in the whole-brain analysis (top panel, STG; middle panel, parieto-occipital cortex; bottom panel, calcarine sulcus).

DOI: https://doi.org/10.7554/eLife.31640.004

frequency was also restricted by the length of the time window used for the analysis. That is, with a 1 s time window and a resulting 1 Hz frequency resolution, a non-integer center frequency at, for example, 6.5 Hz was not feasible. The source-reconstructed coherence at the frequency-range of interest confirmed that cerebro-acoustic coherence was strongest across groups and conditions in bilateral temporal lobes, including primary auditory cortex. (*Figure 1F*).

To test whether envelope tracking in the current study is modulated by intelligibility, we compared the coherence maps for the intelligible (nat) versus non-intelligible (1-channel) condition, in all groups combined (i.e., EB and SI), with dependent-samples permutation t-tests in SPM. The resulting statistical map (*Figure 1G*, p<0.05, FWE-corrected) revealed a cluster in right superior and middle temporal cortex (STG, MTG), where synchronization to the envelope of speech was stronger when participants had access to the content of the story. In other words, the effect of intelligibility on the

**Table 1.** Proportion of correct responses for the story comprehension questions.
Depicted are the number of subjects (N) in the early blind (EB) and sighted individuals (SI) groups, the mean (M), and the standard error of the mean (SE) for each of the three conditions (Natural, 8-channel, 1-channel).

| Group | N | Natural M | Natural SE | 8-channel M | 8-channel SE | 1-channel M | 1-channel SE |
|---|---|---|---|---|---|---|---|
| EB | 17 | 0.899 | 0.019 | 0.853 | 0.031 | 0.559 | 0.027 |
| SI | 15 | 0.914 | 0.020 | 0.890 | 0.025 | 0.576 | 0.034 |

DOI: https://doi.org/10.7554/eLife.31640.003

sensory response (coherence with the speech envelope) suggests an interface between acoustic speech processing and comprehension in STG that is present in both groups.

## Speech tracking in blind parieto-occipital cortex

Whether EB individuals recruit additional neural substrate for tracking the envelope of speech was tested using independent-samples permutation t-tests in SPM, contrasting coherence maps between EB and SI for all three conditions combined. The statistical maps (*Figure 1H*, p<0.05, FWE-corrected) revealed enhanced coherence in EB versus SI in parieto-occipital cortex along the medial wall, centered on bilateral Precuneus, branching more extensively into the right hemisphere (see *Table 2*). This main effect of group highlights that neuronal populations in blind parieto-occipital cortex show enhanced synchronization to the acoustic speech signal. That is, blind participants show a stronger sensitivity to the sensory properties of the external speech stimulus at this level.

Finally, to investigate whether and where envelope tracking is sensitive to top-down predictions during intelligible speech comprehension, we subtracted the unintelligible (1-channel) from the intelligible (nat) condition, and computed independent-samples t-tests between groups. As highlighted in the introduction, we were particularly interested in the role played by the Calcarine sulcus (V1) in processing semantic attributes of speech (*Burton et al., 2002*; *Röder et al., 2002*; *Amedi et al., 2003*), and therefore we restricted the statistical analysis to the area around the primary visual cortex. The search volume was constructed from four 10 mm spheres around coordinates in bilateral Calcarine sulcus ([−7,–81, −3]; [−6,–85, 4]; [12, -87, 0; 10, –84, 6]). The coordinates were extracted from a previous study on speech comprehension in EB (*Burton et al., 2002*). The resulting mask also include regions highlighted in similar studies by other groups (*Röder et al., 2002*; *Bedny et al., 2011*).

A significant effect of the interaction between blindness and intelligibility [(nat$_{EB}$ − 1-channel$_{EB}$) − (nat$_{SI}$ − 1-channel$_{SI}$)] was observed in right calcarine sulcus. To explore the spatial specificity of the effect, we show the whole-brain statistical map for the interaction between intelligibility and blindness at a more liberal threshold (p<0.005, uncorrected) in *Figure 1I*. This shows that intelligible speech selectively engages the area around the right calcarine sulcus in the blind versus the sighted. Specifically, the region corresponding to right 'visual' cortex showed enhanced sensitivity to intelligible speech in EB versus SI (p<0.05, FWE-corrected). However, the analysis contrasting the two intelligible conditions (nat and 8-channel) did not yield a significant effect in the EB, suggesting that the low-level degrading of the stimulus alone does not drive the effect, but rather the intelligibility of the speech segment. Follow-up post hoc comparisons between the two groups revealed that coherence was stronger during the intelligible condition for EB versus SI (t [30] = 3.09, p=0.004), but not during the unintelligible condition (t[30] = –1.08, p=0.29).

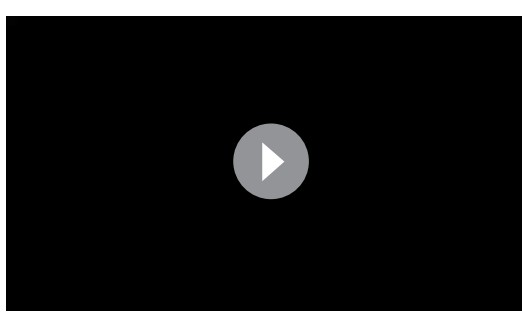

**Video 1.** Movie of the natural and two vocoding conditions used for one exemplar of auditory segment used in our experiment.
DOI: https://doi.org/10.7554/eLife.31640.007

**Table 2.** MNI (Montreal Neurological Institute) coordinates and p-values for the three contrasts tested with threshold-free cluster enhancement (TFCE).

| | | MNI-coordinates (mm) | | |
|---|---|---|---|---|
| | p-value (FWE-cor) | x | y | z |
| Intelligibility | | | | |
| Superior temporal gyrus | 0.027 | 64 | −16 | 0 |
| Blindness | | | | |
| Posterior cingulate | 0.030 | 8 | −40 | 40 |
| Postcentral sulcus | 0.033 | 16 | −32 | 64 |
| Precuneus | 0.033 | -8 | −64 | 48 |
| Blindness x intelligibility | | | | |
| Calcarine sulcus | 0.007 | 16 | −88 | 0 |

DOI: https://doi.org/10.7554/eLife.31640.005

A group difference between EB and SI across conditions was not observed in the calcarine region (t[30] = 1.1, p=0.28). The lack of an overall group effect in calcarine sulcus suggests that there is not a simple enhanced sensory response to speech in the blind. This was different to the response we observed in parietal cortex, in which synchronization was stronger in the blind than in the sighted. Rather, the response in calcarine sulcus only differed between the two groups when speech was intelligible. While overall coherence with the speech envelope was equally high in EB and SI, the blind population showed significantly higher coherence in the intelligible speech condition than did the sighted, who showed higher cerebro-acoustic coherence in the unintelligible condition (1-Chan). The latter is reminiscent of the fact that the more adverse the listening condition (low signal-to-noise ratio or audiovisual incongruence), the more the visual cortex is entrained to the visual speech signal of actual acoustic speech when presented together with varying levels of acoustic noise (*Park et al., 2016*; *Giordano et al., 2017*). Moreover, *Giordano et al. (2017)* showed an increase of directed connectivity between superior frontal regions and visual cortex under the most challenging (acoustic noise and uninformative visual cues) conditions, again suggesting a link between the reorganization observed in the occipital cortex of blind individuals and typical multisensory pathways involving the occipital cortex in audio-visual speech processing (*Kayser et al., 2008*). Visual deprivation since birth, however, triggers a functional reorganization of the calcarine region that can then dynamically interact with the intelligibility of the speech signal. For illustration purposes, cerebro-acoustic coherence from functional peak locations for intelligibility (across groups) in STG, blindness in the parieto-occipital cortex (*Figure 1G*; POC), and the interaction are represented as boxplots in *Figure 1J*.

### Changes in occipito-temporal connectivity in the EB

To further investigate whether cerebro-acoustic peaks in CS and STG which are sensitive to intelligible speech in EB are indicative of a more general re-organization of the network, we conducted functional connectivity analysis. Statistical analysis of the connectivity estimates was performed on the mean phase-locking value in the theta (4–8 Hz) range.

Using linear mixed-effects models, blindness (EB, SI), intelligibility (nat, 1-channel), and the interaction between blindness and intelligibility were added to the model in a hierarchical fashion. Blindness and intelligibility were modeled as fixed effects, while subject was a random effect. Intelligibility was nested in subject. A main effect was observed only for blindness ($\chi$[1] = 4.32, p=0.038) and was caused by greater connectivity for EB versus SI (*Figure 2A–B*). The main effects of intelligibility and of the interaction between intelligibility and blindness were non-significant (p=0.241 and p=0.716, respectively).

Subsequently, linear mixed-effects models were applied to test for the effects of blindness and intelligibility on the directional information flow (PSI) between CS and STG. The fixed and random effects structure was the same as that described in the previous analysis. Here, only the main effect of blindness was significant ($\chi$[1] = 4.54, p=0.033), indicating that the directional information flow differs between groups. The effect of intelligibility and the interaction between intelligibility and

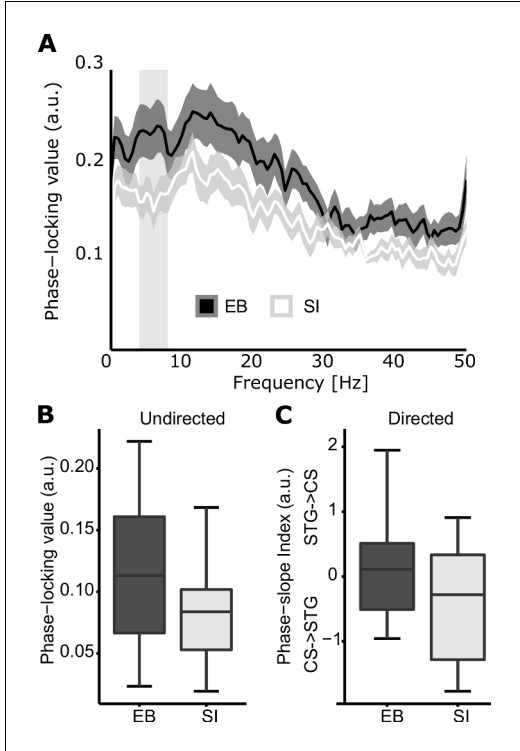

**Figure 2.** Occipital-temporal connectivity in EB and SI. (**A**) Spectra illustrate the phase locking between CS and STG in EB (dark curve) and in SI (light curve). Shaded areas illustrate the SEM. A difference between the curves is observed in the theta range. (**B**) Boxplots depict the mean phase locking between CS and STG for EB (black) and SI (white) in the theta range. Connectivity is enhanced in EB versus SI. (**C**) Directional connectivity analysis using the phase-slope index (PSI). Positive values suggest enhanced directionality from STG to CS, and negative values represent enhanced connectivity in the opposite direction. The boxplots highlight the strong feed-forward drive from CS to STG shown by SI, whereas blind individuals show a more balanced pattern, with a non-significant trend in the opposite direction.

DOI: https://doi.org/10.7554/eLife.31640.006

blindness were both non-significant (p=0.51 and p=0.377, respectively). Follow-up post-hoc one-sample t-tests on the phase slope estimates for each group individually revealed a significant direction bias from CS to STG for SI ($t[14] = -2.22$, p=0.044). No directional bias was found for EB ($t[16] = 0.74$, p=0.47). As depicted in *Figure 2C–D*, these results suggest that CS projects predominantly to STG in SI, whereas in EB, this interaction is more balanced and trending in the opposite direction.

## Discussion

To test whether neuronal populations in the early blind's occipital cortex synchronize to the temporal dynamics present in human speech, we took advantage of the statistical and conceptual power offered by correlating magnetoencephalographic recordings with the envelope of naturalistic continuous speech. Using source-localized MEG activity, we confirm and extend previous research (*Scott et al., 2000*; *Davis and Johnsrude, 2003*; *Narain et al., 2003*; *Rodd et al., 2005*, *2010*; *Okada et al., 2010*; *Peelle et al., 2010*) by showing that temporal brain regions entrain to the natural rhythm of speech signal in both blind and sighted groups (*Figure 1F*). Strikingly, we show that following early visual deprivation, occipito-parietal regions enhance their synchronization to the acoustic rhythm of speech (*Figure 1G*) independently of language content. That is, the region around the Precuneus/Cuneus area seems to be involved in low-level processing of acoustic information alone. Previous studies have demonstrated that this area enhances its response to auditory information after early, but not late, visual deprivation (*Collignon et al., 2013*) and maintains this elevated response to sound even after sight-restoration early in life (*Collignon et al., 2015*). These studies did not find that this crossmodal response was related to a specific cognitive function, but rather pointed

to a more general response to sounds (*Collignon et al., 2013*, *2015*). Interestingly, the current data suggest that this area is capable of parsing complex temporal patterns in sounds but is insensitive to the intelligibility of speech, again suggesting a more general role in sound processing.

In order to isolate the brain regions that are modulated by speech comprehension, we identified regions showing a greater response to amplitude modulations that convey speech information as compared to amplitude modulations that do not. In addition to the right STG observed in both groups (*Figure 1H*), the blind showed enhanced cerebro-acoustic coherence during intelligible speech in the vicinity of calcarine sulcus (V1; *Figure 1I*). This pattern of local encoding was accompanied by enhanced occipito-temporal connectivity during speech comprehension in EB as compared to SI. SI show the expected feed-forward projections from occipital to temporal regions (*Lamme et al., 1998*), whereas EB show a more balanced connectivity profile, trending towards the reverse temporal to occipital direction (see *Figure 2*). These findings support the idea of a reverse hierarchical model (*Büchel, 2003*) of the occipital cortex in EB, where the regions typically coding

for 'low-level' visual features in the sighted (e.g. visual contrast or orientation) participate in higher-level function (e.g. speech intelligibility). Indeed, previous studies have found increased activity in the primary 'visual' cortex of EB people during Braille reading (*Sadato et al., 1996*; *Burton et al., 2002*, *2012*), verbal memory and verb generation tasks (*Amedi et al., 2003*), and during auditory language-related processing (*Bedny et al., 2011*). In line with our results, activity in primary occipital regions in EB people is stronger in a semantic versus a phonologic task (*Burton et al., 2003*), and vary as a function of syntactic and semantic complexity (*Röder et al., 2002*; *Bedny et al., 2011*; *Lane et al., 2015*). Moreover, repetitive transcranial magnetic stimulation (rTMS) over the occipital pole induces more semantic errors than phonologic errors in a verb-generation task in EB people (*Amedi et al., 2003*). As we show that occipital regions entrain to the envelope of speech and are enhanced by its intelligibility, our results clearly suggest that the involvement of the occipital pole for language is not fully abstracted from sensory inputs as previously suggested (*Bedny, 2017*). The role of the occipital cortex in tracking the flow of speech in EB people may constitute an adaptive strategy to boost perceptual sensitivity at informational peaks in language.

## Mechanistic origins of envelope tracking in occipital cortex

Although language is not the only cognitive process that selectively activates the occipital cortex of EB people, it is arguably one of the most puzzling. The reason is that reorganization of other processes, such as auditory motion perception and tactile object recognition, appears to follow the topography of the functionally equivalent visual processes in the sighted brain (*Ricciardi et al., 2007a*; *Amedi et al., 2010*; *Dormal et al., 2016*). For example, the hMT+/V5 complex, which is typically involved in visual motion in the sighted, selectively processes auditory (*Poirier et al., 2006*; *Dormal et al., 2016*; *Jiang et al., 2016*) or tactile (*Ricciardi et al., 2007b*) motion in blind people. However, in the case of language, such recruitment is striking in light of the cognitive and evolutionary differences between vision and language (*Bedny et al., 2011*). This led to the proposal that, at birth, human cortical areas are cognitively pluripotent: capable of assuming a broad range of unrelated cognitive functions (*Bedny, 2017*). However, this argument resides on the presupposition that language has no computational relation with vision. But does this proposition tally with what we know about the relationship between the visual system and the classical language network? Rhythmic information in speech has a well-known language-related surrogate in the visual domain: lip movements (*Lewkowicz and Hansen-Tift, 2012*; *Park et al., 2016*). Indeed, both acoustic and visual speech signals exhibit rhythmic temporal patterns at prosodic and syllabic rates (*Chandrasekaran et al., 2009*; *Schwartz and Savariaux, 2014*; *Giordano et al., 2017*). The perception of lip kinematics that are naturally linked to amplitude fluctuations in speech serves as an important vehicle for the everyday use of language (*Kuhl and Meltzoff, 1982*; *Weikum et al., 2007*; *Lewkowicz and Hansen-Tift, 2012*) and helps language understanding, particularly in noisy conditions (*Ross et al., 2007*). Indeed, reading lips in the absence of any sound activates both primary and association auditory regions overlapping with regions that are active during the actual perception of spoken words (*Calvert et al., 1997*). The synchronicity between auditory and visual speech entrains rhythmic activity in the observer's primary auditory and visual regions, and facilitates perception by aligning neural excitability with acoustic or visual speech features (*Schroeder et al., 2008*; *Schroeder and Lakatos, 2009*; *Giraud and Poeppel, 2012*; *Mesgarani and Chang, 2012*; *Peelle and Davis, 2012*; *van Wassenhove, 2013*; *Zion Golumbic et al., 2013b*; *Park et al., 2016*; *Giordano et al., 2017*). These results strongly suggest that both the auditory and the visual components of speech are processed together at the earliest level possible in neural circuitry, based on the shared slow temporal modulations (around 2–7 Hz range) present across modalities (*Chandrasekaran et al., 2009*). Corroborating this idea, it has been demonstrated that neuronal populations in visual cortex follow the temporal dynamics of lip movements in sighted individuals, similar to the way in which temporal regions follow the acoustic and visual fluctuations of speech (*Luo et al., 2010*; *Zion Golumbic et al., 2013a*; *Park et al., 2016*). Similar to temporal cortex, occipital cortex in the sighted also shows enhanced lip tracking when attention is directed to speech content. This result highlights the fact that a basic oscillatory architecture for tracking the dynamic aspects of (visual-) speech in occipital cortex exists even in sighted individuals.

Importantly, audiovisual integration of the temporal dynamics of speech has been suggested to play a key role when learning speech early in life: young infants detect, match, and integrate the auditory and visual temporal coherence of speech (*Kuhl and Meltzoff, 1982*; *Rosenblum et al.,*

*1997*; *Lewkowicz, 2000, 2010*; *Brookes et al., 2001*; *Patterson and Werker, 2003*; *Lewkowicz and Ghazanfar, 2006*; *Kushnerenko et al., 2008*; *Bristow et al., 2009*; *Pons et al., 2009*; *Vouloumanos et al., 2009*; *Lewkowicz et al., 2010*; *Nath et al., 2011*). For instance, young infants between 4 and 6 months of age can detect their native language from lip movements only (*Weikum et al., 2007*). Around the same period, children detect synchrony between lip movements and speech sounds, and distribute more attention towards the mouth than towards the eyes (*Lewkowicz and Hansen-Tift, 2012*). Linking what they hear to the lip movements may provide young infants with a stepping-stone towards language production (*Lewkowicz and Hansen-Tift, 2012*). Moreover, infants aged 10 weeks already exhibit a McGurk effect, again highlighting the early multisensory nature of speech perception (*Rosenblum et al., 1997*). Taken together, these results suggest that an audio-visual link between observing lip movements and hearing speech sounds is present at very early developmental stages, potentially from birth, which helps infants acquire their first language.

In line with these prior studies, the current results may support the biased connectivity hypothesis of cross-modal reorganization (*Reich et al., 2011*; *Hannagan et al., 2015*; *Striem-Amit et al., 2015*). Indeed, it has been argued that reorganization in blind occipital cortex may be constrained by functional pathways to other sensory and cognitive systems that are also present in sighted individuals (*Elman et al., 1996*; *Hannagan et al., 2015*). This hypothesis may explain the overlap in functional specialization between blind and sighted individuals (*Collignon et al., 2011*; *Dormal et al., 2016*; *He et al., 2013*; *Jiang et al., 2016*; *Peelen et al., 2013*; *Pietrini et al., 2004*; *Weeks et al., 2000*; *Poirier et al., 2006*). In our experiment, sensitivity to acoustic dynamics of intelligible speech in blind occipital cortex could arise from pre-existing occipito-temporal pathways connecting the auditory and visual system that are particularly important for the early developmental stages of language acquisition. In fact, the reorganization of this potentially predisposed pathway to process language content would explain how language-selective response may appear in blind children as young as 3 years old (*Bedny et al., 2015*).

Previous studies have suggested that language processing in the occipital cortex arises through top-down projections from frontal regions typically associated with the classical language network (*Bedny et al., 2011*; *Deen et al., 2015*), and that the representational content is symbolic and abstract rather than sensory (*Bedny, 2017*). Our results contrast with this view by showing that neuronal populations in (peri-)calcarine cortex align to the temporal dynamics of intelligible speech, and are functionally connected to areas sensitive to auditory information in temporal cortex. In sighted individuals, regions of the temporal lobe including STS are sensitive to acoustic features of speech, whereas higher-level regions such as anterior temporal cortex and left inferior frontal gyrus are relatively insensitive to these features and therefore do not entrain to the syllabic rate of speech (*Davis and Johnsrude, 2003*; *Hickok and Poeppel, 2007*; see confirmation in *Figure 1F–G*). This suggests that occipital areas respond, at least partially, to speech at a much lower (sensory) level than previously thought in EB brains, which may be caused by the reorganization of existing multisensory pathways connecting the 'auditory' and 'visual' centers in the brain.

Functional dependencies between sensory systems exist between the earliest stages of sensory processing in both human (*Ghazanfar and Schroeder, 2006*; *Kayser et al., 2008*; *Schroeder and Lakatos, 2009*; *Murray et al., 2016*) and nonhuman primates (*Falchier et al., 2002*; *Lakatos et al., 2007*; *Schroeder and Lakatos, 2009*). Several neuroimaging studies have demonstrated enhanced functional connectivity in sighted individuals between auditory and visual cortices under multisensory conditions (see *Murray et al., 2016* for a recent review), including multisensory speech (*Giordano et al., 2017*). Moreover, neuroimaging studies have shown that hearing people consistently activate left temporal regions during silent speech-reading (*Calvert et al., 1997*; *MacSweeney et al., 2000, 2001*). We therefore postulate that brain networks that are typically dedicated to the integration of audio-visual speech signal, might be reorganized in the absence of visual inputs and might lead to an extension of speech tracking in the occipital cortex (*Collignon et al., 2009*). Although an experience-dependent mechanism related to EB affects the strength and directionality of the connectivity between the occipital and temporal regions, the presence of intrinsic connectivity between these regions — which can also be observed in sighted individuals (e.g. for multisensory integration) — may constrain the expression of the plasticity observed in our task. Building on this connectivity bias, the occipital pole may extend its sensitivity to the intelligibility of speech, a computation this region is obviously not originally dedicated to.

A number of recent studies have suggested that visual deprivation reinforces the functional connections between the occipital cortex and auditory regions typically classified as the language network (*Hasson et al., 2016*; *Schepers et al., 2012*). Previous studies using dynamic causal modelling support the idea that auditory information reaches the reorganized occipital cortex of the blind through direct temporo-occipital connection, rather than using subcortical (*Klinge et al., 2010*) or top-down pathways (*Collignon et al., 2013*). In support of these studies, we observed that the overall magnitude of functional connectivity between occipital and temporal cortex is higher in blind people than in sighted people during natural speech comprehension. Moreover, directional connectivity analysis revealed that the interaction between the two cortices is also qualitatively different: sighted individuals show a strong feed-forward drive towards temporal cortex, whereas blind individuals show a more balanced information flow, and a trend in the reverse direction. These results highlight one possible pathway by which the speech signal is enhanced in the occipital cortex of EB individuals. However, this does not mean that the changes in connectivity between blind and sighted individuals are limited to this specific network (e.g. see *Kayser et al., 2015*; *Park et al., 2015*).

## Speech comprehension in the right hemisphere

We observed that neuronal populations in right superior temporal cortex synchronize to the temporal dynamics of intelligible, but not non-intelligible, speech in both EB and SI groups. Why does speech intelligibility modulate temporal regions of the right, but not the left, hemisphere? According to an influential model of speech comprehension – the asymmetric sampling in time model (AST; *Giraud and Poeppel, 2012*; *Hickok and Poeppel, 2007*; *Poeppel, 2003*) – there is a division of labour between the left- and right auditory cortices (*Poeppel, 2003*; *Boemio et al., 2005*; *Hickok and Poeppel, 2007*), with the left auditory cortex being more sensitive to high-frequency information (+20 Hz), whereas the right temporal cortex is more sensitive to low-frequency information (~6 Hz) such as syllable sampling and prosody (*Belin et al., 1998*; *Poeppel, 2003*; *Boemio et al., 2005*; *Obleser et al., 2008*; *Giraud and Poeppel, 2012*). Several studies have shown that the right hemisphere is specifically involved in the representation of connected speech (*Bourguignon et al., 2013*; *Fonteneau et al., 2015*; *Horowitz-Kraus et al., 2015*; *Alexandrou et al., 2017*), whereas other studies have directly demonstrated the prevalence of speech-to-brain entrainment while listening to sentences or stories in delta and theta bands in the right hemisphere more than in the left hemisphere (*Luo and Poeppel, 2007*; *Abrams et al., 2008*; *Gross et al., 2013*; *Giordano et al., 2017*). The present study therefore replicates these results by showing enhanced phase coupling between the right hemisphere and the speech envelope at the syllabic rate (low-frequency phase of speech envelope), consistent with the AST model. An interesting observation in the current study is that right hemispheric sensitivity to intelligible speech in temporal areas coincides with the enhanced right hemispheric sensitivity to intelligible speech in the occipital cortex of blind individuals.

Having more cortical tissue devoted to sentence processing and understanding could potentially support enhanced sentence comprehension. Previous studies have indeed demonstrated that, as compared to sighted individuals, blind people have enhanced speech discrimination in noisy environments (*Niemeyer and Starlinger, 1981*), as well as the capability to understand speech displayed at a much faster rate (sighted listeners at rates of 9–14 syllables/s and blind listeners at rates of 17–22 syllables/s; *Moos and Trouvain, 2007*). Crucially, listening to intelligible ultra-fast speech (as compared to reverse speech) has been shown to cause enhanced activity in the right primary visual cortex in early and late blind individuals when compared to sighted controls, and activity in this region correlates with individual ultra-fast speech perception skills [*Dietrich et al., 2013*]). These results raise the interesting possibility that the engagement of right V1 in the analysis of the speech envelope, as demonstrated in our study, may support the enhanced encoding of early supra-segmental aspects of the speech signal, supporting the ability to understand an ultra-fast speech signal. However, our behavioral task (speech comprehension) did not allow us to directly assess this link between the reorganized occipital cortex and speech comprehension, as by design performance was almost at ceiling in the nat and 8-chan condition but at chance in the 1-chan condition (see *Table 1*).

However, it is important to note that linking brain activity and behavior is not a trivial issue. Many studies have not found a direct link between crossmodal reorganization and non-visual processing.

More generally, as a behavioral outcome is the end product of a complex interactive process between several brain regions, linking the role of one region in isolation (e.g. the reorganized region in the occipital cortex of blind people) to behavior (e.g. performance) in a multifaced task is not straightforward. An absence of a direct relation between behavior (e.g. speech processing) and crossmodal plasticity could be explained by the fact that this complex process is supported by additional networks other than the reorganized ones. Interestingly, recent studies have shown that early visual deprivation triggers a game of 'balance' between the brain systems typically dedicated to a specific process and the reorganized occipital cortex (*Dormal et al., 2016*). It has indeed been proposed that early visual deprivation triggers a redeployment mechanism that would reallocate part of the processing typically implemented in the preserved networks (i.e. the temporal or frontal cortices for speech processing) to the occipital cortex deprived of its most salient input (vision). Two recent studies using multivoxel pattern analysis (MVPA) showed that the ability to decode the different auditory motion stimuli was enhanced in hMT+ (a region typically involved in visual motion in sighted individuals) of early blind individuals, whereas an enhanced decoding accuracy was observed in the planum temporale in the sighted group (*Jiang et al., 2016*; *Dormal et al., 2016*). Moreover, *Bedny et al. (2015)* reported an enhanced activation of occipital cortex and a simultaneous deactivation of prefrontal regions during a linguistic task in blind children. The authors suggested that the increased involvement of the occipital cortex might decrease the pressure on the prefrontal areas to specialize in language processing. Interestingly, a transcranial magnetic stimulation (TMS) study reported a reduced disruptive effect of TMS applied over the left inferior prefrontal cortex during linguistic tasks for both blind and sighted individuals, whereas TMS applied over the occipital cortex caused more disruption in early blind as compared to sighted people (*Amedi et al., 2004*). Such studies support the idea that brain regions that are typically recruited for specific tasks in sighted individuals may become less essential in EB people if they concomitantly recruit occipital regions for the same task. An important open question for future research therefore concerns the relative behavioral contribution of occipital and perisylvian cortex to speech understanding.

## Conclusions

We demonstrate that the primary 'visual' cortex synchronizes to the temporal dynamics of intelligible speech at the rate of syllable transitions in language (~6–7 Hz). Our results demonstrate that this neural population is involved in processing the sensory signal of speech, and therefore contrasts with the proposition that occipital involvement in speech processing is abstracted from its sensory input and purely reflects higher-level operations similar to those observed in prefrontal regions (*Bedny, 2017*). Blindness, due to the absence of organizing visual input, leaves the door open for sensory and functional colonization of occipital regions. This colonization might however not be stochastic, but could be constrained by modes of information processing that are natively anchored in specific brain regions and networks. Even if the exact processing mode is still to be unveiled by future research, we hypothesise that the mapping of language onto occipital cortex builds upon pre-existing oscillatory architecture typically linking auditory and visual speech rhythm (*Chandrasekaran et al., 2009*). Our study therefore supports the view that the development of functional specialisation in the human cortex is the product of a dynamic interplay between genetic and environmental factors during development, rather than being predetermined at birth (*Elman et al., 1996*).

## Materials and methods

### Participants

Seventeen early blind (11 female; mean ± SD, 32.9 ± 10.19 years; range, 20–67 years) and sixteen sighted individuals (10 female; mean ± SD, 32.2 ± 9.92 years; range, 20–53 years) participated in the current study. There was no age difference between the blind and the sighted group ($t$[30] = 0.19, p=0.85). All blind participants were either totally blind or severely visually impaired from birth; however, two participants reported residual visual perception before the age of 3 and one before the age of 4, as well as one participant who lost their sight completely at age 10. Causes of vision loss were damage or detached retina (10), damage to the optic nerve (3), infection of the eyes (1), microphtalmia (2), and hypoxia (1). Although some participants reported residual light perception, none

were able to use vision functionally. All participants were proficient braille readers and native speakers of Italian. None of them suffered from a known neurological or peripheral auditory disease. The data from one sighted individual were not included because of the discovery of a brain structural abnormality that was unknown to the experimenters at the time of the experiment. The project was approved by the local ethical committee at the University of Trento. In agreement with the Declaration of Helsinki, all participants provided written informed consent before participating in the study.

## Experimental design

Auditory stimuli were delivered into the magnetically shielded MEG room via stereo loudspeakers using a Panphonics Sound Shower two amplifier at a comfortable sound level, which was the same for all participants. Stimulus presentation was controlled via the Matlab (RRID:SCR_001622) Psychophysics Toolbox 3 (http://psychtoolbox.org; RRID:SCR_002881) running on a Dell Alienware Aurora PC under Windows 7 (64 bit). Both sighted and blind participants were blindfolded during the experiment, and the room was dimly lit to allow for visual monitoring of the participant via a video feed from the MEG room. Instructions were provided throughout the experiment using previous recordings from one of the experimenters (FB).

The stimuli consisted of 14 short segments (~1 min) from popular audiobooks (e.g., Pippi Longstocking and Candide) in Italian (nat). Furthermore, channel-vocoding in the Praat software was used to produce two additional control conditions. First, the original sound file was band-pass filtered into 1 (1-channel) or 8 (8-channel) logarithmically spaced frequency bands. The envelope for each of these bands was computed, filled with Gaussian white noise, and the different signals were recombined into a single sound file. The resulting signal has an amplitude envelope close to the original, while the fine spectral detail was gradually distorted. Perceptually, the voice of the speaker is highly distorted in the 8-channel condition, but intelligibility is unperturbed. By contrast, in the 1-channel condition, speech is entirely unintelligible. In total, 42 sound files were presented in a pseudo-randomized fashion, distributed among seven blocks. Each block contained two sound files from each condition, and the same story was never used twice in the same block. Examples of the stimuli are provided in supplementary media content ( see *Video 1*).

To verify story comprehension, each speech segment was followed by a single-sentence statement about the story. Participants were instructed to listen to the story carefully, and to judge whether the statement at the end was true or false, using a nonmagnetic button box. Responses were provided with the index and middle finger of the right hand.

## MEG data acquisition and preprocessing

MEG was recorded continuously from a 306 triple sensor (204 planar gradiometers; 102 magnetometers) whole-head system (Elekta Neuromag, Helsinki, Finland) using a sampling rate of 1 kHz and an online band-bass filter between 0.1 and 300 Hz. The headshape of each individual participant was measured using a Polhemus FASTRAK 3D digitizer. Head position of the subject was recorded continuously using five localization coils (forehead, mastoids).

Data pre-processing was performed using the open-source Matlab toolbox Fieldtrip (www.fieldtriptoolbox.org; RRID:SCR_004849). First the continuous data were filtered (high-pass Butterworth filter at 1 Hz, low-pass Butterworth filter at 170 Hz, and DFT filter at 50, 100, and 150 Hz to remove line-noise artefacts in the signal), and downsampled to 256 Hz. Next, the data were epoched into segments of 1 s for subsequent analysis.

Rejection of trials containing artefacts and bad channels was performed using a semi-automatic procedure. First, outliers were rejected using a pre-screening based on the variance and range in each trial/channel. Then, algorithmically guided visual inspection of the raw data was performed to remove any remaining sources of noise.

## Extraction of the speech amplitude envelope

Amplitude envelopes of the stories were computed using the Chimera toolbox (*Chandrasekaran et al., 2009*) and custom code, following the procedure described by Gross and colleagues (*Gross et al., 2013*). First, the sound files were band-pass filtered between 100 and 1000 Hz into nine frequency-bands, using a fourth order Butterworth filter. The filter was applied in forward and backward direction to avoid any phase shifts with respect to the original signal, and

frequency-bands were spaced with equal width along the human basilar membrane. Subsequently, the analytic amplitude for each filtered segment was computed as the absolute of the Hilbert transform. Finally, the amplitude envelopes of all nine bands were summed and scaled to a maximum value of 1. The resulting envelope was combined with the MEG data, and processed identically henceforth.

## Analysis of cerebro-acoustic coherence

To determine where, and at what temporal scale, neuronal populations follow the temporal dynamics of speech, we computed spectral coherence between the speech envelope and the MEG signal. Coherence is a statistic that quantifies the phase relationship between two signals, and can be used to relate oscillatory activity in the brain with a peripheral measure such as a speech signal (*Peelle et al., 2013*). The first analysis was conducted in sensor space, across conditions and participants, to determine the temporal scale at which coherence between the speech envelope and the MEG signal is strongest. To this end, a Hanning taper was applied to the 1 s data segments, and Fourier transformation was used to compute the cross-spectral density between 1 and 30 Hz, with a step size of 1 Hz. To render the coherence values more normally distributed, a Fisher z-transform was applied by computing the inverse hyperbolic tangent (atanh).

Source-space analysis was centered on the coherence frequency-band of interest (FOI) identified in the sensor space analysis. Source reconstruction was performed using a frequency domain beamformer called Dynamic Imaging of Coherent Sources (DICS) (*Gross et al., 2001*; *Liljeström et al., 2005*). DICS was used to localize spectral coherence, observed at the sensors, on a three-dimensional grid (8 × 8 × 8 mm). The forward model was based on a realistic single-shell headmodel (*Nolte, 2003*) for each individual. As structural scans could not be acquired for all participants, we approximated individual anatomy by warping a structural MNI template brain (MNI, Montreal, Quebec, Canada; www.bic.mni.mcgill.ca/brainweb) into individual headspace using the information from each participant's headshape.

## Source connectivity analysis

Functional connectivity between occipital and temporal cortex was computed by extracting virtual sensor time-series at the locations of interest in CS and STG using time-domain beamforming. These virtual sensor time series were used to compute non-directional and directional connectivity metrics. The rationale behind focusing our analyses on STG and CS is to limit our hypothesis space by the results as they unfolded in our analytic steps. Indeed, we found that the right STG was the only region modulated by the intelligibility of speech in both groups, confirming previous results using similar methods (*Luo and Poeppel, 2007*; *Abrams et al., 2008*; *Gross et al., 2013*; *Giordano et al., 2017*). In addition to STG, we found enhanced cerebro-acoustic coherence for intelligible speech in CS of EB individuals. We therefore decided to focus our connectivity analyses on these two ROIs since they were functionally defined from our first analytical step (modulation of cerebro-acoustic coherence by speech intelligibility). The advantage of following this procedure is that connectivity analyses are based on nodes that we can functionally interpret and for which we have clear predictions. We successfully used the same hierarchical analytic structure in previous studies (*Collignon et al., 2013*, *2015*; *Benetti et al., 2017*), following guidelines on how to investigate directional connectivity in brain networks (*Stephan et al., 2010*).

Virtual sensor time-series at the two locations of interest were extracted using a time-domain vector-beamforming technique called linear constrained minimum variance (LCMV) beamforming (*Van Veen et al., 1997*). First, average covariance matrices were computed for each participant to estimate common spatial filter coefficients. These filter coefficients were multiplied with the single-trial cleaned MEG data. To reduce the resulting three-dimensional time-series to one singular value, decomposition (SVD) was applied, resulting in a single time-series for each trial and participant.

Functional connectivity between the two regions was computed at peak locations using the phase-locking value (PLV) (*Lachaux et al., 1999*). First, single-trial virtual sensor time courses were converted to the frequency domain (0–50 Hz) using Fourier transformation. The data were padded to 2 s and a Hanning taper was applied to reduce spectral leakage. Coherence was used as a proxy for functional connectivity. To disentangle phase consistency between regions from joint fluctuations in power, the spectra were normalized with respect to the amplitude, resulting in an estimate of the

phase-locking (*Lachaux et al., 1999*) between regions. Connectivity estimates were normalized using a Fischer z-transform (atanh), as in the analysis of cerebro-acoustic coherence.

PLV is a symmetric proxy for connectivity and does not allow for inferences regarding the direction of information flow between two regions of interest. To test whether differences in functional connectivity between EB and SI are accompanied by changes in the directionality of the flow of information between the regions of interest, we computed the phase slope index (PSI) (*Nolte et al., 2008*). The PSI deduces net directionality from the time delay between two time series (x1 and x2). Time series x1 is said to precede, and hence drive, time series x2 in a given frequency band if the phase difference between x1 and x2 increases with higher frequencies. Consequently, a negative phase slope reflects a net information flux in the reverse direction, that is, from x2 to x1. Here, we computed the PSI using a bandwidth of ±5 Hz around the frequencies of interest. Following the recommendation by Nolte and colleagues (*Nolte et al., 2008*), PSI estimates were normalized with the standard error, which was computed using the jackknife method.

### Statistical analysis

Statistical testing of the behavioral comprehension scores, as well as the connectivity estimates, was performed using linear mixed-effects models in R. Differences between conditions and groups in source space were evaluated using Statistical Parametric Mapping (SPM12; RRID:SCR_007037), and the Threshold-Free Cluster Enhancement (TFCE) toolboxes in Matlab. TFCE (*Smith and Nichols, 2009*) computes new values for each voxel in a statistical map as a function of the original voxel value and the values of the surrounding voxels. By enhancing the statistical values in voxels with a high T-value that are also part of a local cluster, TFCE optimally combines the benefits of voxel-wise and cluster-based methods. TFCE was applied to the whole-brain contrasts. Final correction for multiple comparisons was applied using a maximum statistic based on 1000 permutations of group membership (independent testing) or conditions (dependent testing). Here, we applied a variance smoothing of 15 mm FWHM to reduce the effects of high spatial frequency noise in the statistical maps. The smoothing kernel used in the current study is higher than that in comparable fMRI studies because of the inherent smoothness of the source-reconstructed MEG data.

## Acknowledgements

The project was funded by the ERC grant MADVIS – Mapping the Deprived Visual System: Cracking function for Prediction (Project: 337573, ERC-20130StG) awarded to Olivier Collignon. We would also like extend our gratitude to Valeria Occelli, and to the Masters students who assisted with the data acquisition (Marco Barilari, Giorgia Bertonati, and Elisa Crestale) for their kind assistance with the blind participants. In addition, we would like to thank Gianpiero Monittola for ongoing support with the hardware and Jodie Davies-Thompson for insightful comments on the manuscript. Finally, we would like to thank the organizations for the blind in Trento, Mantova, Genova, Savona, Cuneo, Torino, Trieste and Milano for their help in recruiting participants.

## Additional information

### Funding

| Funder | Grant reference number | Author |
|---|---|---|
| H2020 European Research Council | 337573 | Markus Johannes Van Ackeren<br>Stefania Mattioni<br>Roberto Bottini<br>Olivier Collignon |

The funders had no role in study design, data collection and interpretation, or the decision to submit the work for publication.

### Author contributions

Markus Johannes Van Ackeren, Conceptualization, Data curation, Software, Formal analysis, Supervision, Validation, Investigation, Visualization, Methodology, Writing—original draft, Writing—review

and editing; Francesca M Barbero, Conceptualization, Data curation, Validation, Methodology, Project administration, Writing—review and editing; Stefania Mattioni, Roberto Bottini, Resources, Data curation, Validation, Investigation, Project administration, Writing—review and editing; Olivier Collignon, Conceptualization, Resources, Data curation, Supervision, Funding acquisition, Validation, Visualization, Methodology, Writing—original draft, Project administration, Writing—review and editing

## Author ORCIDs
Francesca M Barbero 🔟 https://orcid.org/0000-0003-4445-4402
Roberto Bottini 🔟 http://orcid.org/0000-0001-7941-7762
Olivier Collignon 🔟 http://orcid.org/0000-0003-1882-3550

## Ethics
Human subjects: The project was approved by the local ethical committee at the University of Trento (protocol 2014-007). In agreement with the Declaration of Helsinki, all participants provided written informed consent to participate in the study.

## Decision letter and Author response
Decision letter https://doi.org/10.7554/eLife.31640.010
Author response https://doi.org/10.7554/eLife.31640.011

## Additional files

### Supplementary files
• Transparent reporting form
DOI: https://doi.org/10.7554/eLife.31640.008

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
