## [Decision Letter]

Thank you for submitting your article "Neuronal populations in the occipital cortex of the blind synchronize to the temporal dynamics of speech" for consideration by *eLife*. Your article has been reviewed by three peer reviewers, and the evaluation has been overseen by a Reviewing Editor and Andrew King as the Senior Editor. The following individuals involved in review of your submission have agreed to reveal their identity: Elana Zion Golumbic (Reviewer #2); Yanchao Bi (Reviewer #3).

The reviewers have discussed the reviews with one another and the Reviewing Editor has drafted this decision to help you prepare a revised submission.

Summary:

This paper examines the role of the deprived visual cortex in the processing of language in blind individuals. Using magnetoencephalography, the authors report that, compared to individuals who could see, early-blind subjects show more synchronized brain responses within the calcarine sulcus to the temporal dynamics of speech. Intriguingly, they observed enhanced synchronization to the intelligibility of speech in the region of the primary visual cortex, and changes in the flow of information between areas of the occipital and temporal cortex that are sensitive to speech intelligibility. The reviewers agreed that this is an important topic and that this is a timely study that builds nicely on previous work on the neural tracking of speech and in the domain of language-system plasticity in the blind. However, various issues were identified with the analysis and presentation of the data, which have implications for the conclusions made by the authors.

Essential revisions:

1) The authors offer up a compelling narrative, but do not address the fundamental question of why blind participants need to recruit visual cortices for speech comprehension. What is the added functional value of having a more expanded brain network? The manuscript provides no explanation for this central aspect of the results. This needs to be addressed in the Discussion.

2) The reviewers found your description of the data analysis confusing and inadequate. Please be more specific about exactly which group or interaction effects are used as the marker for sensory processing. If this is based on the interaction between group and speech intelligibility in the calcarine sulcus, as assumed by one of the reviewers, other contrasts, e.g., the main effect of group (do blind and sighted subjects differ in terms of the coherence effects?) should also be reported. Given that the three conditions were matched on speech envelope properties, wouldn't interaction with intelligibility actually reflect top down modulation, which is presumably from lexical/semantic/syntactic properties rather than sensory processes?

3) There are several conflicting sentences in the Results, which make it unclear whether synchronization is present in both sighted and early-blind individuals, with the two groups differing in the degree of tracking, or is present only in the early blind. From Figure 1 it seems that the sighted did in fact show both a general coherence effect (at least for the 8 and 1 channel conditions) and an intelligibility effect (negative correlation). Demonstrating clearly whether the temporal locking effects are present in the sighted subjects and how exactly the early-blind group behaves in comparison is important because this directly links to the conclusions drawn in the paper. It appears from this figure that the functional role of the calcarine sulcus in synchronizing to speech in these populations may need reconsideration.

4) The reviewers pointed out the analysis of source reconstructions on two separate frequency windows that were largely overlapping needs clearer explanation and justification.

5) Another issue concerns the presentation of the same story clip in 3 different versions to each participant. Again, this requires clearer explanation and justification.

---

## [Author Response]

Summary:This paper examines the role of the deprived visual cortex in the processing of language in blind individuals. Using magnetoencephalography, the authors report that, compared to individuals who could see, early-blind subjects show more synchronized brain responses within the calcarine sulcus to the temporal dynamics of speech. Intriguingly, they observed enhanced synchronization to the intelligibility of speech in the region of the primary visual cortex, and changes in the flow of information between areas of the occipital and temporal cortex that are sensitive to speech intelligibility. The reviewers agreed that this is an important topic and that this is a timely study that builds nicely on previous work on the neural tracking of speech and in the domain of language-system plasticity in the blind. However, various issues were identified with the analysis and presentation of the data, which have implications for the conclusions made by the authors.Essential revisions:1) The authors offer up a compelling narrative, but do not address the fundamental question of why blind participants need to recruit visual cortices for speech comprehension. What is the added functional value of having a more expanded brain network? The manuscript provides no explanation for this central aspect of the results. This needs to be addressed in the Discussion.

Why crossmodal plasticity arises in the brain of blind individuals is a crucial question that the field is currently trying to address. It is a very complex question, however, since the underlying mechanisms governing the expression of crossmodal plasticity is the result of a complex interplay between intrinsic organizational principles and experience-dependent influences on these intrinsic neurophysiological constraints.

One possibility that has been proposed in the literature is that the reorganization observed in the occipital cortex of blind people would support enhanced perceptual abilities for the reorganized function (De Villers-Sidani et al., 2016). The rational is simple: since an extension in the cortical territories underlying a computation is observed in the blind, this extension may support enhanced compensatory behavior. For instance, it has been demonstrated that early blind individuals with the higher auditory localization abilities (Gougoux et al., 2005) or superior verbal memory (Amedi et al., 2003) also show higher occipital activity levels. Moreover, occipital cortical thickness predicts performance on pitch and musical tasks in blind individuals (Voss and Zatorre, 2012). Similarly, in early deaf humans the higher the recruitment of temporal regions for face processing the better they are at processing faces (Benetti et al., 2017) and temporarily deactivating the functioning of crossmodally reorganized regions of the temporal cortex of deaf cats also alleviates their superior performance in visuo-spatial processing (Lomber et al., 2010). All-together these studies suggest that the crossmodal recruitment of sensory deprived regions may induce enhanced abilities in the remaining senses.

In relation to the topic of the current study, having more cortical tissue devoted to sentence processing and understanding could potentially support enhanced sentence comprehension. Previous studies have indeed demonstrated that blind people have enhanced speech discrimination in noisy environments (Niemeyer and Starlinger, 1981) and the capability to understand speech displayed at a much faster rate than sighted persons (sighted listeners at rates between 9-14 syllables/s and to blind listeners between 17-22 syllables /s; (Moos and Trouvain, 2007) Crucially, listening to intelligible ultra-fast speech (compared to reverse speech) engage enhanced activity in the right primary visual cortex in early and late blind individuals when compared to sighted controls; and activity in this region correlated with individual ultra-fast speech perception skills (Dietrich et al., 2013). These results raise the interesting possibility that the engagement of right V1 in the analysis of the speech envelope as demonstrated in our study may support the enhanced encoding of early supra-segmental aspects of the speech signal, supporting the ability to understand ultra-fast speech signal. Our behavioral task (speech comprehension) did not however create enough between-subject variability to assess directly this link between the reorganized occipital cortex and speech comprehension (performance was almost at ceiling in the Nat and 8-chan condition but at chance in the 1-chan condition, as targeted- see Table 1).

However, it is important to note that linking brain activity and behavior is not a trivial issue. Many studies did not find a direct link between crossmodal reorganization and non-visual processing. More generally, since a behavioral outcome is the end product of a complex interactive process between several brain regions, linking the role of one region in isolation (e.g. the reorganized region in the occipital cortex of blind people) to behavior (e.g. performance) in a multifaced task is not straightforward. The absence of a direct relation between behavior, e.g. speech processing, and crossmodal plasticity could be explained, for instance, by the fact that this complex process will be supported by additional networks than the reorganized ones. Interestingly, recent studies have shown that early visual deprivation triggers a game of “balance” between the brain systems typically dedicated to a specific process and the reorganized occipital cortex (Dormal et al., 2016). It has indeed been proposed that early visual deprivation triggers a redeployment mechanism that would reallocate part of the processing typically implemented in the preserved networks (i.e. the temporal or frontal cortices for speech processing) to the occipital cortex deprived of its most salient input (vision). Two recent studies using multivoxel pattern analysis (MVPA) showed that the ability to decode the different auditory motion stimuli was enhanced in hMT+ (a region typically involved in visual motion in sighted) of early blind while an enhanced decoding accuracy was observed in the sighted group in the planum temporale (Dormal et al., 2016; Jiang et al., 2016). Moreover, Bedny and collaborators (2015) reported enhanced activation of occipital cortex and a simultaneous deactivation of prefrontal regions during a linguistic task in blind children. The authors suggested that the increased involvement of the occipital cortex might decrease the pressure on the prefrontal areas to specialize for language. Interestingly, a Transcranial Magnetic Stimulation (TMS) study reported a reduced disruptive effect of TMS applied over the left inferior prefrontal cortex during linguistic tasks, while TMS application over the occipital had more effects in early blind compared to sighted people (Amedi et al., 2004). This evidence supports the idea that brain regions normally recruited for specific tasks in sighted might become less essential in EB in the case they concomitantly recruit occipital regions for the same task. An important open question for future research therefore concerns the relative behavioral contribution of occipital and perisylvian cortex to speech understanding.

We now have summarized these arguments in the revised Discussion section of our manuscript.

2) The reviewers found your description of the data analysis confusing and inadequate. Please be more specific about exactly which group or interaction effects are used as the marker for sensory processing. If this is based on the interaction between group and speech intelligibility in the calcarine sulcus, as assumed by one of the reviewers, other contrasts, e.g., the main effect of group (do blind and sighted subjects differ in terms of the coherence effects?) should also be reported. Given that the three conditions were matched on speech envelope properties, wouldn't interaction with intelligibility actually reflect top down modulation, which is presumably from lexical/semantic/syntactic properties rather than sensory processes?

We thank the reviewers for pointing out that our previous description of the data analysis was not clear enough. We now have thoroughly modified our data analysis section in order to make it more streamlined. We clarify here, in brief, the specific points raised by the reviewers.

The marker for sensory processing of speech is the cerebro-acoustic coherence that is independent of the intelligibility of speech. Therefore, the regions that are solely sensitive to the acoustic information of our signal (e.g. variation of acoustic energy across time, at the syllabic rate) are tracking the signal envelope for the unaltered speech segments but also for the two vocoded conditions, including the 1-chanel vocoding condition during which speech is not intelligible. Figure 1 therefore shows the regions that synchronize most strongly to the acoustic properties of speech independently of the group (sighted and blind) and independently of the condition of presentation (Natural speech, 8 channels vocoding and 1 channel vocoding). Figure 1 however shows regions where the cerebro-acoustic coherence is higher in the blind than in the sighted, but still independently of condition presentation. This is the effect of group. These regions therefore reorganize in the blind to track acoustic stimulation but do not necessarily show an effect of intelligibility of the speech signal. It is certainly the case that phase-locked responses to sensory stimuli are not unique to human speech comprehension, as evidenced by oscillatory responses to simple auditory stimuli in non-human primates (Lakatos et al., 2007; Schroeder et al., 2008). Might there be additional information in the speech signal that affects the phase-locked cortical response to spoken language in the occipital cortex of the blind? This is exactly what we target with the group-level effect of blindness. The interaction effect we observed in calcarine sulcus (CS) highlights that synchronization to intelligible speech is different in the blind versus sighted. (see Figure 1). As such, the enhanced neural phase-locking observed in the calcarine sulcus of early blind individuals is not solely driven by changes in the acoustic cue of the auditory stimuli, but also reflects cortical encoding and processing of the speech signal.

As we now discuss at length in the manuscript, the calcarine sulcus in the blind might therefore locate at the interface between high-level speech comprehension and acoustic processing. This sheds important new lights on language processing in the occipital cortex of the blind by demonstrating that the involvement of this neural population relates to the sensory signal of speech and therefore contrasts with the dominant proposition that occipital involvement in speech processing is abstracted from its sensory input and purely reflect higher-level operations similar to those observed in prefrontal regions (Bedny, 2017)

3) There are several conflicting sentences in the Results, which make it unclear whether synchronization is present in both sighted and early-blind individuals, with the two groups differing in the degree of tracking, or is present only in the early blind. From Figure 1 it seems that the sighted did in fact show both a general coherence effect (at least for the 8 and 1 channel conditions) and an intelligibility effect (negative correlation). Demonstrating clearly whether the temporal locking effects are present in the sighted subjects and how exactly the early-blind group behaves in comparison is important because this directly links to the conclusions drawn in the paper. It appears from this figure that the functional role of the calcarine sulcus in synchronizing to speech in these populations may need reconsideration.

We thank the reviewers for this valuable question. With the evidence presented in the current study we are describing differences in the degree of cerebro-acoustic tracking. A binary classification as to whether tracking does, or does not occur in one of the groups was not intended. We outline our reasons for this argument below.

First, from a practical point of view, coherence is a bivariate metric that ranges between 0 and 1. Thus, with no negative values, a simple one-sample t-test in this case is not possible. Indeed, the majority of studies use group or condition comparisons to identify regions where cerebro-acoustic tracking is modulated. Here we use the contrast between intelligible and vocoded speech to highlight areas sensitive to speech intelligibility, and the effect of blindness on target areas where sensitivity to the speech rhythm. Hence all our conclusions are limited to describing modulatory effects.

Second, while it is theoretically possible to compare the results with a surrogate distribution to mimic a one-sample t-test, we believe that this analysis would not add to our argument. The current study was designed to address questions regarding group and condition differences. Additional one-sample t-test would not affect these conclusions. For example, if we do find a significant difference from zero, the conclusions would still hold, while a lack of an effect could not be interpreted. That said, our main proposal is based on the idea that the blind recycle existing functional pathways, and therefore we would not be surprised to find weak forms of cerebro-acoustic coherence in the occipital cortex of the sighted as well. For example, following the reviewer’s suggestion, we now have also examined the group effect as a post-hoc comparison in CS. Here we do not find that coherence is overall (across the 3 conditions) higher in blind versus sighted individuals. However, we do find a qualitative difference where the blind show enhanced coherence during intelligible speech compared to sighted individuals, while in the unintelligible condition, we find no significant difference between groups. The fact that the occipital cortex of sighted individuals shows stronger entrainment for altered speech condition may relate to the previous demonstration that the more adverse the listening condition (low signal-to-noise-ratio or audio-visual incongruence), the more the visual cortex is entrained to the speech signal of actual acoustic speech presented together with varying levels of acoustic noise (Park et al., 2016; Giordano et al., 2017). Moreover, Giordano and colleagues (2017), showed an increase of directed connectivity between superior frontal regions and visual cortex under the most challenging (acoustic noise and uninformative visual cues) conditions. Kayser et al. (Kayser et al., 2015) also proposed top-down processes modulating acoustic entrainment. In case of absence of structuring visual inputs since birth, the occipital cortex that does not play a role in integrating visual and auditory speech signals, like in the sighted, would now start to show enhanced coherence during intelligible speech, like what is typically observed in temporal regions. This hypothesis again suggests a link between the reorganization observed in blind individuals and the typical multisensory structure involving the occipital cortex in audio-visual speech processing (Kayser et al., 2007).

We now have added the following paragraph to the Results section to highlight that idea.

“A group difference between blind and sighted individuals across conditions was not observed in the calcarine region (t(30)=1.1, p=.28). The lack of an overall group effect in calcarine sulcus suggests that there is not a simple enhanced sensory response to speech in the blind. In this respect, calcarine sulcus responds differently than the region wide identified in parietal cortex where synchronization was overall stronger in the blind. Rather, the two groups differ only when speech is intelligible. But, while overall coherence with the speech envelope is equally high in early blind and sighted individuals, the blind population show significantly higher coherence in the intelligible speech condition compared to the sighted, who show higher cerebro-acoustic coherence in the unintelligible condition (1-Chan). The latter is reminiscent to the fact that the more adverse the listening condition (low signal-to-noise-ratio or audio-visual incongruence), the more the visual cortex is entrained to visual speech signal of actual acoustic speech presented together with varying levels of acoustic noise (Park et al., 2016; Giordano et al., 2017). Moreover, Giordano and colleagues (2017), showed an increase of directed connectivity between superior frontal regions and visual cortex under the most challenging (acoustic noise and uninformative visual cues) conditions. This hypothesis again suggests a link between the reorganization observed in the occipital cortex of blind individuals and typical multisensory pathways involving the occipital cortex in audio-visual speech processing (Kayser et al., 2007). Visual deprivation since birth however triggers a functional reorganization of the calcarine region that may now dynamically interact with the intelligibility of speech signal.”

4) The reviewers pointed out the analysis of source reconstructions on two separate frequency windows that were largely overlapping needs clearer explanation and justification.

We apologize for this lack of clarity. We have decided to combine the reconstructions with filters for two different frequency bands to optimally capture the shape of the coherence spectrum we saw in sensors space. We have decided to opt for this approach to cover the broad range of frequencies where syllables can occur. In addition, our choice was restricted by our decision to segment the trials into 1s segments, and hence compute a frequency spectrum with integer frequencies. A center-frequency at 6.5Hz was thus not feasible. We have now elaborated on this approach in the Materials and methods.

By combining the two frequency-bands we acquire a source estimate that emphasizes the center of our frequency band of interest (6-7Hz) and tapers off towards the edges, which optimally represents the coherence spectrum observed in sensor space. The choice of the center frequency was also restricted by the length of the time window used for the analysis. That is, with a 1s time window and a resulting 1Hz frequency resolution, a non-integer center frequency at e.g. 6.5Hz was not feasible.

5) Another issue concerns the presentation of the same story clip in 3 different versions to each participant. Again, this requires clearer explanation and justification.

Because the key analyses involved contrasting the different conditions, we wanted to make sure that the speech envelopes for the three different conditions were exactly the same and, so it was crucial to keep the sentences the same. However, to avoid contamination as much as possible, we ensured that during the randomization procedure of the trials two versions of the same story were never presented close to each other. Due to the length and complexity of the stories presented in the current study, it would be very difficult to reconstruct a story in the 1chan condition from a previously heard comprehensible version. We argue that this is a different situation than hearing short sentences in close succession during a vocoded and intelligible condition. In addition, if this effect existed, it would be balanced out due to the randomization procedure.